# A systematic screen identifies Saf5 as a link between splicing and transcription in fission yeast

Sonia Borao�, Montserrat Vega☉�, Susanna Boronat☉�, Elena Hidalgo☉,
Stefan Hümmer☉¤*, José Ayté☉*

Oxidative Stress and Cell Cycle Group, Universitat Pompeu Fabra, Barcelona, Spain

య These authors contributed equally to this work.
¤ Current address: Translational Molecular Pathology, Vall d'Hebron Research Institute (VHIR) and CIBERONC, Barcelona, Spain
* stefan.hummer@vhir.org (SH); jose.ayte@upf.edu (JA)

## Abstract

Splicing is an important step of gene expression regulation in eukaryotes, as there are many mRNA precursors that can be alternatively spliced in different tissues, at different cell cycle phases or under different external stimuli. We have developed several integrated fluorescence-based *in vivo* splicing reporter constructs that allow the quantification of fission yeast splicing *in vivo* on intact cells, and we have compared their splicing efficiency in a wild type strain and in a *prp2-1* (U2AF65) genetic background, showing a clear dependency between Prp2 and a consensus signal at 5' splicing site (5'SS). To isolate novel genes involved in regulated splicing, we have crossed the reporter showing more intron retention with the *Schizosaccharomyces pombe* knock out collection. Among the candidate genes involved in the regulation of splicing, we have detected strong splicing defects in two of the mutants -Δ*cwf12*, a member of the NineTeen Complex (NTC) and Δ*saf5*, a methylosome subunit that acts together with the survival motor neuron (SMN) complex in small nuclear ribonucleoproteins (snRNP) biogenesis. We have identified that strains with mutations in *cwf12* have inefficient splicing, mainly when the 5'SS differs from the consensus. However, although Δ*saf5* cells also have some dependency on 5'SS sequence, we noticed that when one intron of a given pre-mRNA was affected, the rest of the introns of the same pre-mRNA had high probabilities of being also affected. This observation points Saf5 as a link between transcription rate and splicing.

## Author summary

Diversity in gene expression extends to various levels, and splicing is no exception. Unlike more straightforward organisms like budding yeast, which feature a fundamental set of splicing components and a limited count of constitutively-processed introns, fission yeast stands out. In fission yeast, there is a greater abundance of introns, and the splicing machinery is more intricate, encompassing non-essential splicing factors. In this study,

**Data Availability Statement:** The data generated in the RNA-seq is available in GEO (GSE242312). Original data has been deposited in Figshare (https://doi.org/10.6084/m9.figshare.2591920.v2).

**Funding:** This work was supported by grants BFU2018-PGC2018-097248-B-I00 and PID2022-136449NB-I00 funded by MICIU/AEI/10.13039/501100011033 and ERDF/EU to JA and by Unidad de Excelencia Maria de Maeztu (CEX2018-000792-M) to JA and EH. EH is a recipient of an ICREA Academia Award (Generalitat de Catalunya). The funders had no role in study design, data collection and analysis, decision to publish, or preparation of the manuscript.

**Competing interests:** The authors have declared that no competing interests exist.

we conducted a genetic screening to identify non-essential genes regulating basal splicing. Notably, Saf5 emerged as a standout candidate. Cells lacking Saf5 exhibited significantly impaired growth, with viability limited to 30˚C, indicating heightened sensitivity to both cold and heat stress. A detailed analysis of splicing efficiency revealed that Saf5's role in splicing is not dictated by *cis* elements but is intricately tied to the transcription rate of individual genes. Consequently, Saf5 proves indispensable for the proper splicing of highly transcribed genes, yet dispensable for those with lower transcription rates. This pivotal role positions Saf5 as a crucial link bridging splicing and transcription processes.

## Introduction

Splicing of messenger RNA precursors (pre-mRNA) is a crucial process in the regulation of gene expression. Most pre-mRNA transcripts contain non-coding sequences, known as introns, which must be removed to produce functional mRNA. Splicing involves excising the introns from pre-mRNA and ligating the coding sequences (exons) to form mature mRNA. This process is catalyzed by the spliceosome through two consecutive trans-esterification reactions [1]. The spliceosome's assembly occurs after defining the exon-intron borders during the formation of the earliest spliceosome precursor (E-complex) [2–4].

The definition of the E-complex involves different cis-elements within the pre-mRNA sequence and several splicing factors (trans-elements). The 5' splicing site (ss) is recognized by base pairing with the 5'-end of U1 snRNP [5–7], while the non-snRNP splicing factor SF1 recognizes the Branch Point (BP) [8, 9]. Both subunits of the U2AF heterodimer (U2AF65 and U2AF35) are recruited to the polypyrimidine track and the AG dinucleotide within the 3'ss, respectively [10–13].

Correct splicing is of utmost importance, as its misregulation has been implicated in various human diseases, including cancer, cardiovascular and neurological disorders, diabetes, and Alzheimer's [14]. Disruptions in pre-mRNA splicing leading to human diseases can be categorized into two groups: cis-acting mutations that impede proper recognition of pre-mRNA by splicing factors, modifying the prevalence of constitutive or alternative splice sites, and trans-acting mutations that directly affect the expression of splicing factors, impacting basal or constitutive splicing, or factors regulating alternative splicing, such as SR proteins or hnRNPs [15].

*Schizosaccharomyces pombe*, with approximately 6000 genes, presents an ideal model to study conserved cellular processes across eukaryotes [16,17]. Roughly 43% of its genes contain introns, often multiple ones (up to 15 introns). In contrast to other unicellular yeast species like *Saccharomyces cerevisiae*, which have a limited number of introns, *S. pombe* provides an excellent opportunity to explore the molecular basis of splicing. Notably, fission yeast introns are relatively small compared to budding yeast, with average intron sizes of 83 and 256 nucleotides (nt), respectively. In contrast, in *S. pombe*, the 5'ss and BP exhibit variability comparable to corresponding sequences in mammals [18–20].

Previous studies, including intron lariat sequencing, short-read RNA sequencing (RNA-seq), and Iso-Seq, have unveiled many low-frequency alternative isoforms, indicating relatively low splicing fidelity in fission yeast [21–23]. However, regulated alternative splicing has not been observed as a major issue in mitotically active fission yeast cells [24]. Interestingly, certain genes seem to be regulated by splicing during sexual differentiation. For instance, the 80 nt intron in the pre-mRNA encoding the meiotic cyclin Rem1 is spliced after the initiation of meiosis when the gene is transcribed by the Forkhead family transcription factor Mei4 [25,26].

To search for global regulators of splicing, we have conducted an in vivo screening using the model organism *S. pombe*. To facilitate this study, we developed reporter strains allowing us to measure splicing efficiency through quantitative fluorescence. These reporters are derived from the *rhb1* gene, incorporating its first intron (rhb11-170), with both wild-type and mutated versions. This specific construct, previously used to study the sensitivity of certain introns to Prp2 inactivation (fission yeast U2AF65) [27], will help to gain a deeper understanding of splicing, its regulatory mechanisms and its potential linkage to various diseases.

## Results

### Generation of a genetically-encoded splicing reporter

To explore novel factors involved in the regulation of splicing, we generated a construct containing part of the *rhb1* gene (the first exon, the first intron and the first 49 nucleotides of the second exon). This intron is constitutively spliced in a wild type background, since it has canonical 5'ss and 3'ss [27]. Our engineered reporter was flanked by two different fluorescent proteins (mRFP and YFP), so that only when spliced, yellow fluorescence could be detected since the intron introduces a STOP codon in the ORF (Figs 1A and 1B and S1A). We hypothesized that it would be difficult to find in fission yeast viable deletions of splicing factors capable of regulating the splicing of fully consensus introns. Thus, we generated a set of reporters in which either the 5' splicing site (5'ss), the branchpoint (BP) or both were mutated (5'SS mut, BP mut or dmut, respectively), partially impairing splicing, as described before [27]. Additionally, we also generated two control constructs: one devoid of the intron, serving as a hypothetical 100% splicing scenario (cDNA reporter), and another containing a Stop codon at the start of the YFP (STOP), representing the readout of no splicing, as depicted in Fig 1B. To validate the functionality of the different constructs, we transformed a wild type strain and determined their fluorescence output through microscopic analysis. As shown in Fig 1B, all the constructs reflected the splicing efficiency that we expected.

To ascertain the feasibility of obtaining quantitative or semi-quantitative splicing measurements using our various reporters, we introduced them into a strain where splicing could be conditionally compromised. We used a strain carrying a temperature sensitive allele of *prp2* (U2AF65), *prp2-1* [28,29] in which at restrictive temperature, splicing is impaired in those introns that do not have strictly canonical splicing sequences [27]. Cultures with the different integrative plasmids were grown at the permissive temperature (25°C) or subjected to a 5-hour incubation at the semi-restrictive temperature (30°C), which did not affect the expression of the WT or cDNA reporter (S1B Fig). Then, we measured red and yellow fluorescence of the different cultures by analyzing them by Fluorescence-Activated Cell Sorting (FACS). As shown in Fig 1C, the partial inactivation of Prp2 affected all the constructs except the one lacking an intron between the two fluorescent markers (cDNA). The constructs containing the wild type intron or a mutation in the BP showed a slightly, but consistent reduction of the YFP/RFP ratio (12% and 8%, respectively; Fig 1C) when Prp2 was partially inactivated at 30°C. The most pronounced impact was observed in those constructs in which the last nucleotide preceding the 5'SS was mutated (G-to-U, 5'SS mut reporter). This effect was further increased when the BP was also mutated (dmut reporter).

### Genome-wide screening of non-essential genes to identify genes involved in the regulation of general splicing

As an initial step before embarking on a genome-wide screening and as a proof of concept, we generated a knock-out collection of non-essential genes involved in RNA metabolism from 41

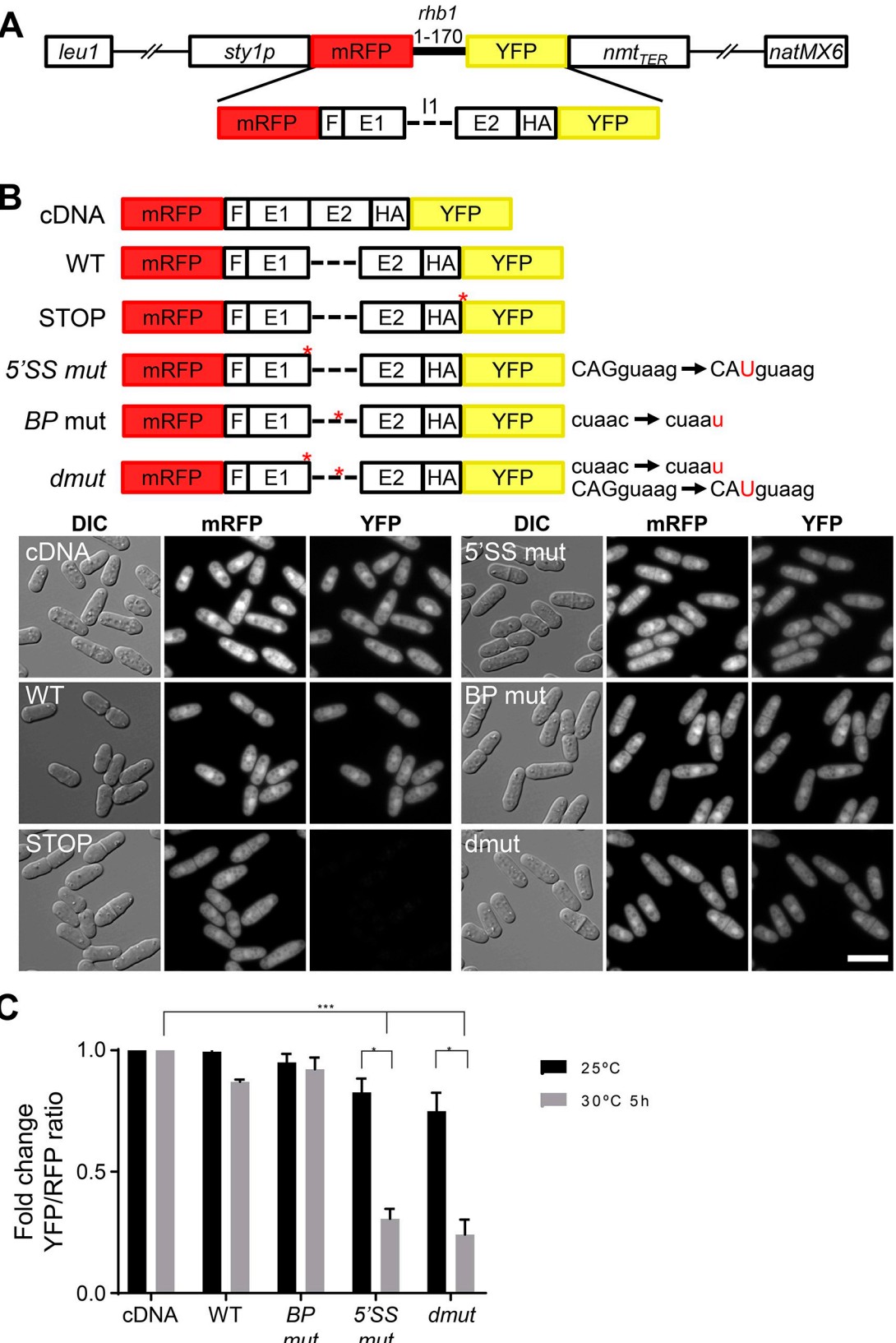

**Fig 1.** (**A**) Schematic representation of the constructs for the in vivo quantification of splicing. *rhb1* intron 1 and both flanking exons are inserted between RFP and YFP and preceded by the epitope HA. *sty1* promoter drives the expression of the construct, which is inserted at the *leu1* locus. natMX6 is inserted at the end of the plasmid for antibiotic selection. (**B**) Six reporters with different mutations, in 5's and BP sequence, were used. WT reporter, non-intron reporter (cDNA), and STOP reporter were also used for *in vivo* splicing quantification. Microscopy images show the DIC, mRFP and YFP fluorescence of cells expressing the indicated reporters. Scale bar, 5 μm. (**C**) *prp2-1* strains were grown at 25°C and at 30°C during 5h. Significant differences were calculated using two-sided t-test (* p<0.05, *** p<0.001).

strains that we already had in the laboratory (S1 Table). This collection was transformed with the integrative WT and the dmut reporters. The transformants were grown in a single 96-well plate at 30°C, and we performed flow cytometry to quantify red and yellow fluorescence for each well. Triplicates of this experiment are represented in Fig 2A, where YFP/RPF ratio is showed in the Y axis. The WT reporter showed minimal variation among the 41 knock out strains, where almost all the strains had a splicing efficiency similar to the WT control strain (shown as a red dot). Conversely, the dmut reporter showed higher variability in the splicing efficiency across the different strains, with increased accumulation of unspliced product (indicated as lower YFP/RFP ratio). Among the 41 tested strains, three showed a markedly low YFP/RFP ratio when compared to the wild type strain: *Δcwf12*, *Δsaf5* and *Δsaf1*. To confirm that the low YFP/RFP ratio was indeed due to a defect on splicing, we conducted RT-PCR analysis in these three strains, which revealed the accumulation of the unspliced form of the dmut reporter (Fig 2B). To further investigate whether the splicing defect observed in the dmut reporter was caused by the mutation in the BP or in the 5'ss, we transformed the strains with reporters carrying individual mutations on the BP (BP mut) or 5'ss (5'ss mut) and measured the YFP/RFP ratio by cytometry. As shown in Fig 2C, the splicing defects in the dmut reporter had the lowest YFP/RFP ratio among all the reporters. The combination of the two mutations (5'ss mut and BP mut) led to an additive effect, with the primary splicing defects predominantly attributed to the 5'ss mutation. Additionally, *Δcwf12* consistently displayed the lowest YFP/RFP ratio across all reporters used (Fig 2B and 2C).

This proof-of-concept study using 41 selected strains showed that our reporter system was efficient to isolate non-essential splicing factors affecting the splicing of *rhb1*. Given that the dmut reporter was able to enhance the defects in splicing and it was more sensitive to small changes in splicing efficiency, we decided to use dmut reporter in a genome-wide screening. Consequently, we crossed the Bioneer collection (approx. 3400 genes) [17] with a reporter strain with de dmut construct (Fig 2D). Analysis of the screening allowed us to identify 37 candidates potentially involved in splicing regulation (S2 Table). Of these candidates, only 4 were already known to be involved in the regulation of splicing (Cwf12, Saf5, Cwf19 and SPAC1705.02), whereas the remaining 33 were unrelated or not clearly assigned to the regulation of splicing. Among the 37 candidates, we decided to confirm the effect on splicing of some of them, including *ΔSPAC1705.02*, a homolog of human 4F5S, and *Δmpn1*, a poly(U)-specific exoribonuclease [30]. As shown in Fig 2E, while the effect detected in the *ΔSPAC1705.02* strain was mild and not statistical significant, the effect observed in the *Δmpn1* strain was comparable to that observed in *Δsaf5* cells.

## Characterization of *Δsaf5* and *Δcwf12* deletions and their role in splicing

Based on the results of our two screenings, we chose to focus on the role of Cwf12 and Saf5 in the regulation of splicing. Cwf12 (Isy1 in budding yeast) forms part of the Prp19-containing NineTeen Complex (NTC). It has been shown that Isy1 acts together with U6 snRNA favoring a spliceosomal conformation before the first splicing reaction. Moreover, *Δisy1* cells show a decrease in the accuracy of 3'ss recognition [31]. Saf5 (Lot5 in *S. cerevisiae*) is described to be

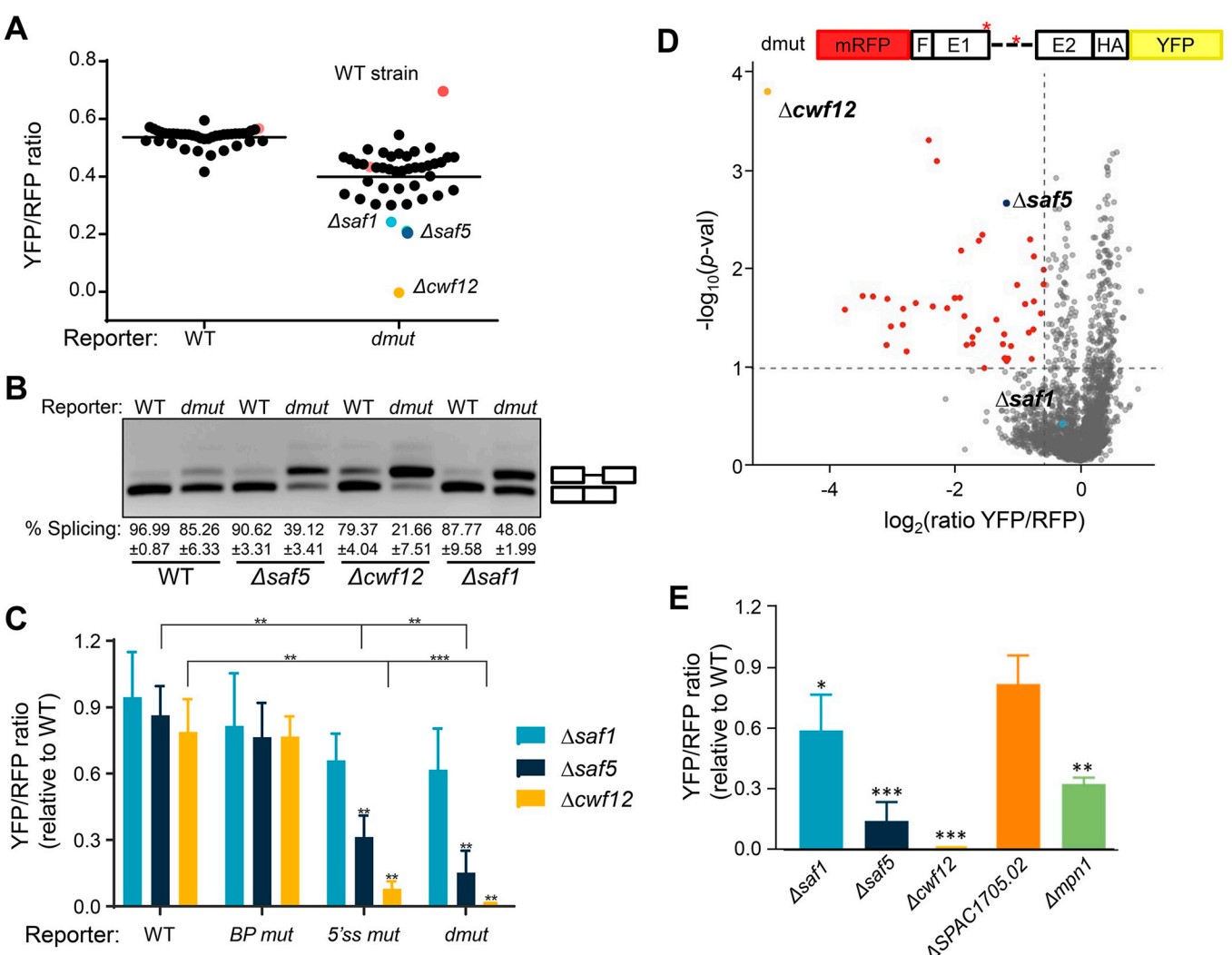

**Fig 2.** (**A**) Triplicates of the 41 selected deletion mutants (S1 Table) were crossed with both reporters (WT and *dmut*). Dots represent the mean of YFP/RFP ratios of each strain. Red dots represent control WT strains with the WT reporter or *dmut* reporter. Blue, dark blue, and yellow dots represent statistically significant values, corresponding to *Δsaf1*, *Δsaf5* and *Δcwf12* strains, respectively. Horizontal lines indicate the mean of all WT/*dmut* wells. (**B**) RT-PCR products showing splicing efficiency in WT and *dmut* reporters in WT, *Δsaf1*, *Δsaf5* and *Δcwf12* strains. Mobility of unspliced and spliced products are indicated on the right. The % of splicing was calculated from 3 independent experiments. (**C**) Fold change of the YFP/RFP ratio values of WT, BP mut, 5'ss mut and dmut reporters in *Δsaf1*, *Δsaf5* and *Δcwf12* strains. The YFP/RFP ratio was normalized to the ratio in the WT strain. Significant differences were calculated using two-sided t-test (** p<0.01) *** p<0.001). (**D**) Volcano plot depicting screening results. Each dot represents a mutant. The x-axis shows the log2 of the YFP/RFP ratio and the y-axis the -log10 of the *p-value*. In red color are marked the significant candidates and in blue, dark blue and yellow color are highlighted the *Δsaf1*, *Δsaf5* and *Δcwf12* strains, respectively. (**E**) Verification of 5 candidates from (**D**) analyzed by flow cytometry in separately grown cultures. Fold change of the YFP/RFP ratio of each strain is normalized to the WT strain (WT = 1). Only the results of two independent experiments in *dmut* construct is shown. Significant differences were calculated using a two-sided t-test (* p<0.05, ** p<0.01 and *** p<0.001).

required for snRNP production and its deletion affects to a subset of introns whose polypyrimidine track is upstream of the BP and their A/U content is lower when compared to normally spliced introns [32]. To further characterize the role of these two genes in the regulation of general splicing, we extracted total RNA from *S. pombe* cells (WT, *Δcwf12* and *Δsaf5*) growing at log-phase. Subsequently, we conducted RNAseq of biological duplicates of each strain to determine the effect of each deletion on general splicing. Our analysis revealed that deletion of *cwf12* had a minor impact on splicing, with only a small amount of introns being affected in their splicing, when compared to a wild type strain. When we categorized the splicing

efficiency of all introns into deciles, the profile of the *Δcwf12* strain in the first five deciles resembled that of the wild-type strain (Fig 3A), especially when compared with that of *Δsaf5* (see below). Similarly, when comparing the splicing efficiency of individual introns in the wild type and *Δcwf12* strains, we observed a strong linear correlation (S2A Fig). This was in contrast to the situation when an essential splicing gene, such as *prp2* (the homologue of human U2AF65), was inactivated, leading to a significant proportion of introns that were not spliced (S2A Fig) [27]. On the contrary, when *saf5* was deleted we observed a more substantial impact on splicing. Classification of splicing efficiency into deciles revealed a greater disturbance in splicing in the *Δsaf5* strain (Fig 3A), especially in deciles 6 to 10. Similarly, when we compared the splicing efficiency of individual introns in *Δsaf5* to those in the WT strain, we also observed a more pronounced impact on splicing, compared to the *Δcwf12* strain (S2A Fig). To ascertain whether the splicing efficiency in the *Δcwf12* or *Δsaf5* strains depended on specific intron characteristics, we analyzed the 5'SS, the 3'SS, and the BP of the 5% most affected introns in these strains (238 introns). However, we could not detect any significant difference with the consensus sequence of all fission yeast introns (Figs 3B and S2B). We also sought to determine potential differences in the intron size (Fig 3C) or the A/T content (Fig 3D), but we found no substantial variations in size or nucleotide composition.

## Saf5 is required for the splicing of highly transcribed genes

*Δsaf5* cells are described to have problems of growth at high temperatures and they are highly sensitive to many drugs [32,33]. When we grew *Δsaf5* cells on serial dilutions on solid media, we noticed that they were temperature sensitive as well as cold sensitive (Fig 4A). In liquid rich media (YE5S), they grew slower than a WT strain, reaching lower saturation (Fig 4B). When we analyzed the transcriptome of *Δsaf5* cells, we noticed that the deletion of *saf5* had a profound impact on the expression of many genes (Fig 4C), with more than 350 genes whose expression was induced over two-fold. Notably, when we did the same analysis with the transcriptome of the *Δcwf12* cells, we only observed the induction of the expression of 150 genes. When we analyzed the Gene Ontology enrichment of the genes whose expression was up-regulated in *Δsaf5* cells, it caught our attention that several meiotic genes were upregulated (Fig 4D), including some genes that are normally not expressed at all in mitotically growing cells, like *rec10*, *bqt1* and *meu13*. These genes belong to a group of genes whose mRNA is recognized by Mmi1 and targeted for degradation during mitotic growth, effectively preventing the presence of their encoded proteins. Interestingly, *mmi1* itself has 4 introns whose splicing is affected to varying degrees in *Δsaf5* cells (74%, 71%, 95% and 85%, respectively). The cumulative effect on the final mRNA would result in only 44% of mature mRNA with the correct splicing in all four introns. To test this hypothesis, we introduced a copy of tagged *mmi1* with GFP in a wild type and a *Δsaf5* background. As shown in S3A Fig, the amount of Mmi1 was significantly reduced in *Δsaf5* cells to less than 50% (S3B Fig), when compared to a wild type strain. On the other hand, replacing the wild type *mmi1* gene with a copy without introns (sfGFP-Mmi1 cDNA), nearly abolished the effect of the absence of Saf5 on the amount of Mmi1 (S3A and S3B Fig).

Finally, and to confirm whether the misregulation of *mmi1* pre-mRNA splicing directly contributed to the increased accumulation of some meiotic transcripts, we quantified the mRNAs of *rec10*, *bqt1* and *meu13* by Q-PCR. We compared a wild type strain with a *Δsaf5* strain expressing *mmi1* without introns and without any tag. As shown in S3C Fig, the strain that was expressing *mmi1* without introns (Mmi1 cDNA) displayed normal levels of *rec10*, *bqt1* and *meu13*. We confirmed by Q-PCR that the strain expressing *mmi1* without introns (in a wild type or in *Δsaf5* background) had just an extra copy of *mmi1* (S3D Fig).

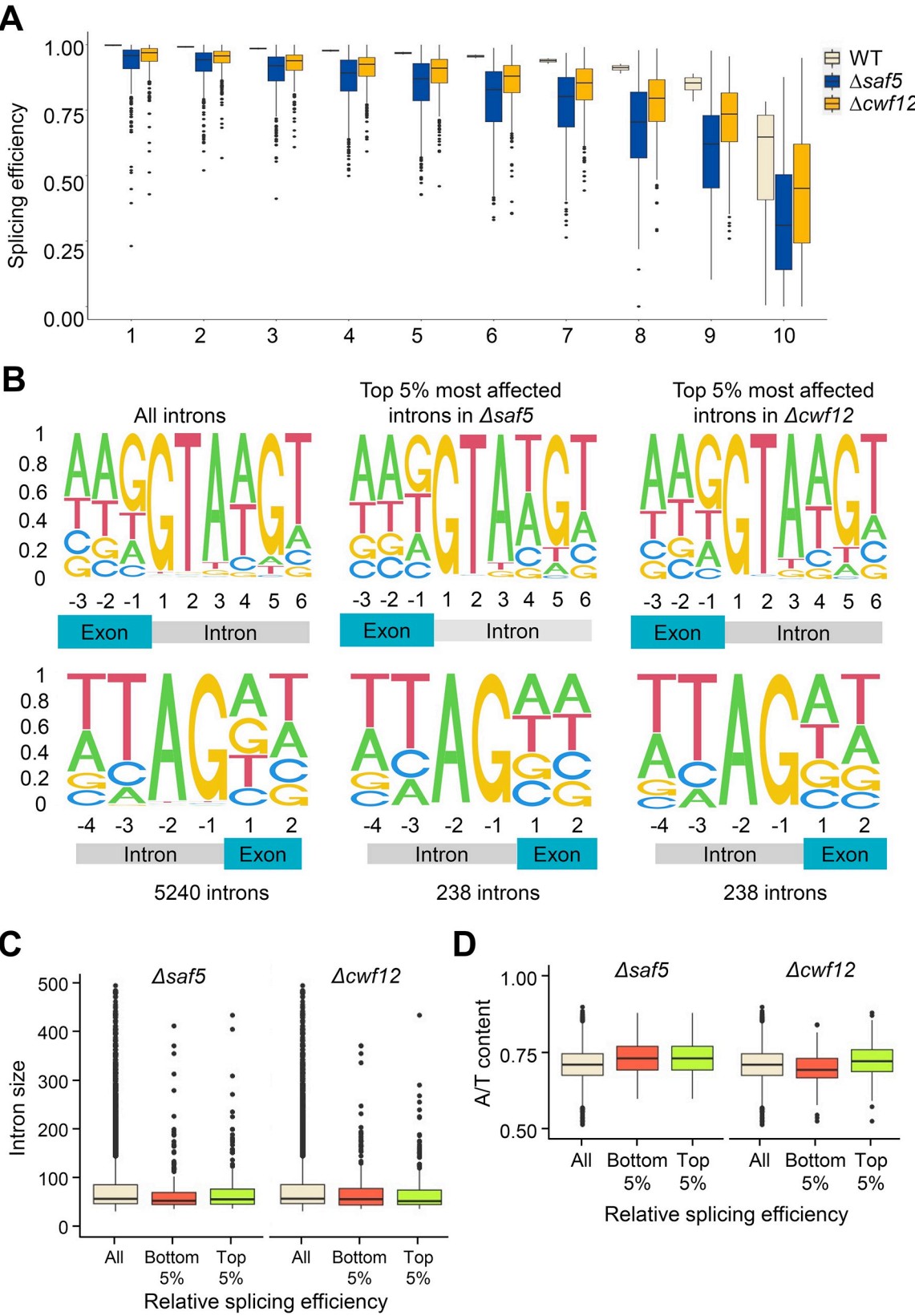

**Fig 3.** (**A**) Splicing efficiency of introns (y-axis) in the wild-type strain (beige) was classified in deciles (x-axis). The boxplot compares the splicing efficiency of these introns in WT (beige), *Δsaf5* (dark blue) and *Δcwf12* (orange) deletion strains. Significant differences were calculated using a two-sided t-test comparing the values of splicing efficiency of each decile in *Δsaf5* or *Δcwf12* strains with the corresponding decile in the wild type strain. All comparisons were statistically significant (p<0.001). (**B**) DNA Sequence of the 5' exon-intron junction (upper panels) and the 3' intron-exon junction (lower panels) of all introns (first column), the 5% worst spliced introns in *Δsaf5* (second column) or *Δcwf12* (third column) strains. (**C**) Intron size of all introns (beige), the 5% worst spliced (red) and the best spliced (green) in a *Δsaf5* and *Δcwf12* strains. (**D**) A/T content of all introns (beige), the 5% worst spliced (red) and the best spliced (green) in a *Δsaf5* and *Δcwf12* strains.

Among all the parameters that we analyzed to determine if there was a common track in those introns whose splicing was affected in the absence of Saf5 (Fig 3), we only found one characteristic that applied to the affected genes: the higher the transcription of a gene, the more likely the splicing of its introns were affected (Fig 4E). If this hypothesis held true, Saf5 should be required similarly in all the introns of poly-intronic genes with high transcription levels. We further examined the splicing efficiency of genes with 3 introns, focusing only on those genes whose first intron displayed less than 50% splicing in a *Δsaf5* strain when compared to a wild type strain. We then determined the splicing efficiency of the remaining two introns (Fig 4F, left panel). We conducted the same analysis for genes where the second (middle panel) or third (right panel) intron exhibited less than 50% splicing. As shown in Fig 4F, when one intron of a poly-intronic pre-mRNA was affected in the *Δsaf5* strain, there were high probabilities that the other introns were similarly affected. Similar analysis was performed for genes containing 3 introns whose splicing was not affected in *Δsaf5* cells (ratio of splicing efficiency *Δsaf5*/WT > 0.9) (S4A Fig) or for genes with low expression and whose expression is not induced in *Δsaf5* cells (S4B Fig). As shown, we could not observe any effect on splicing in either case.

To conclusively demonstrate that Saf5 had an important role in the splicing of highly transcribed genes, independently of intron characteristics, we decided to determine how splicing was affected when transcription was slowed down. We used a *cdk9* allele (*cdk9^as^*) that could be inactivated in the presence of the ATP analog 3-MB-PP1 [34]. Cdk9 is a P-TEFb-associated CDK that phosphorylates Ser2 in the CTD repeats of RNA polymerase II and Spt5, a DSIF transcription elongation factor. These phosphorylations are required for transcription elongation. Therefore, when Cdk9 was inactivated, transcription was slowed down [34,35]. We analyzed the splicing of the single intron of *cbf11*, which is one of the most affected in the *Δsaf5* strain, with 60% of intron retention. The addition of the bulky ATP analog barely affected intron retention in a wild type or a *Δsaf5* strains (Fig 5A). However, upon adding 3-MB-PP1 to the culture of the *cdk9^as^ Δsaf5* strain, splicing of the *cbf11* intron was noticeably improved, with intron retention reduced to 25%. Splicing also improved in the *cdk9^as^ saf5+* strain (depicted by the pale orange bars), probably as the result of decreasing transcription rate and giving more time for the splicing to occur; however, this improvement was not statistically significant. When we analyzed how inactivation of Cdk9 affected the splicing of multi-intronic genes, such as *atg5* or *urm1* (Fig 5B and 5C), we consistently observed improvements in all introns 30 minutes after adding 3-MB-PP1. Interestingly, in the multi-intronic genes, we observed an improvement of splicing in the *cdk9^as^ saf5+* strain, which was statistically significant in 3 of the introns that were tested (*atg5*i2, *atg5*i3 and *urm1*i3). To validate the association between transcription elongation and Saf5-mediated splicing, we opted for an alternative approach not reliant on the inhibition of Ser2 phosphorylation in the C-terminal domain (CTD) repeats of RNA Polymerase II. By treating both wild-type and *Δsaf5* cells with Actinomycin D, a known transcription elongation inhibitor [36], we successfully replicated the findings previously observed in the *cdk9^as^* strain, where the splicing of all the introns that we tested

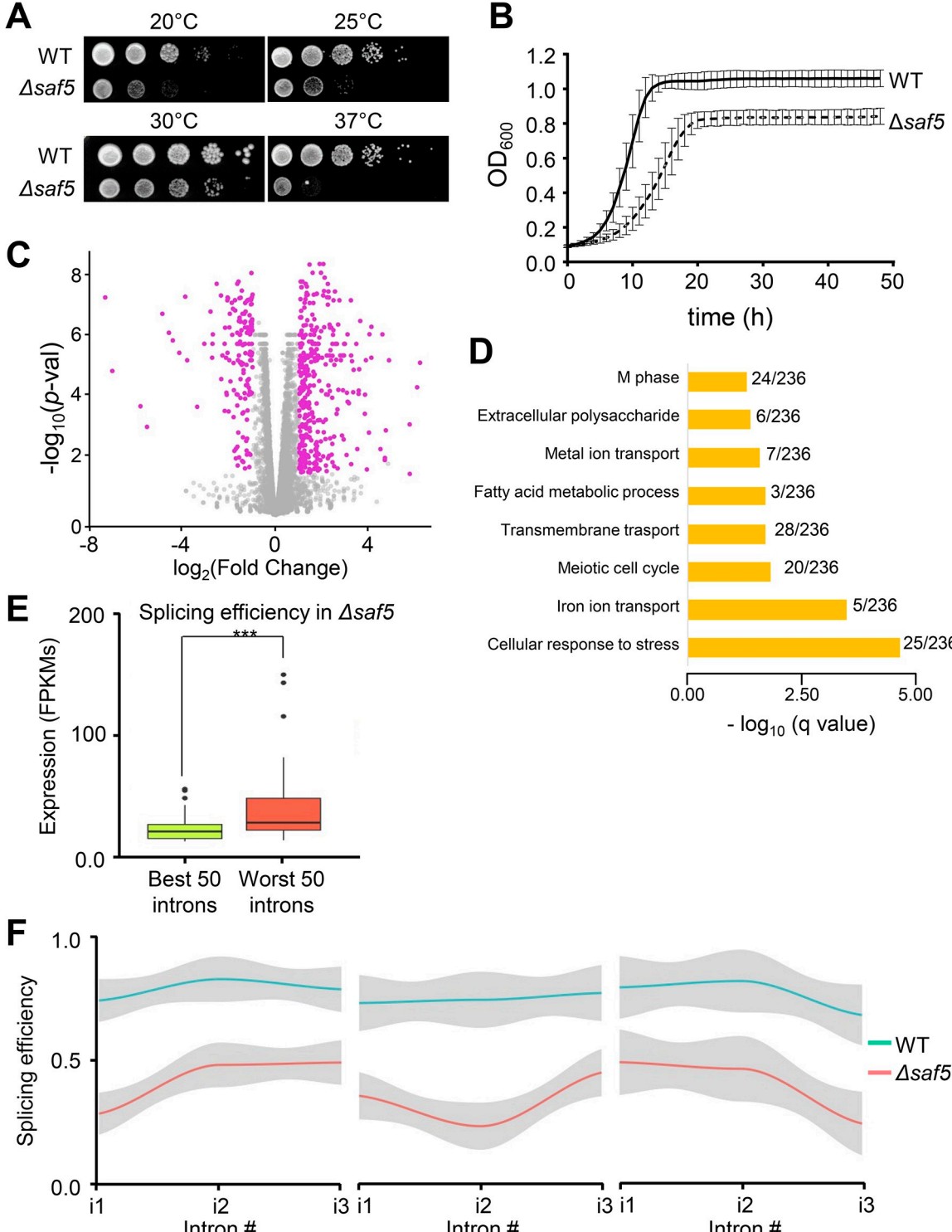

**Fig 4.** (**A**) Growth of WT and *Δsaf5* strains at 18˚C, 25˚C, 30˚C and 37˚C plated on YE5S plates. One representative experiment is shown. (**B**) Growth of WT and *Δsaf5* strains at 30˚C in liquid YE5S media during 48 hours measured by OD600 (y-axis). The average of three independent experiments is shown. Error bars represent the SD. (**C**) Expression volcano plot of *Δsaf5* strain compared to the wild-type strain. Each dot of the scatter plot represents a gene. The x- axis represents the log2 fold-change expression in *Δsaf5* strain compared to the wild-type strain; the y-axis represents the -log10 of the p-value. Magenta dots show the statistically significant genes (log2 fold-change ≥1 and ≤ -1, with a p value < 0.05) (**D**) Gene ontology enrichment. Statistically significant genes from (**C**) were selected for GO enrichment analysis of biological process. Bar plot represents the -log10 of the p-value of the indicated terms. Since

some GO term are redundant, only relevant non-redundant categories are shown. (**E**) Comparison between splicing efficiency and expression levels. Boxplot represents the expression levels (FPKMs) of the 50 best or the worst spliced introns in *Δsaf5* strain (*** p<0.001). (**F**) Splicing efficiency along genes containing three introns. Line plots represent the average splicing efficiency in *Δsaf5* (red line) and wild type (blue line) strains. Genes with a decrease of 50% in splicing efficiency in *Δsaf5* compared with wild type strain in the first (left panel), second (middle panel) and third (right panel) intron are represented.

was noticeably improved (S5 Fig). This observation confirmed our hypothesis that slowing transcription could enhance splicing of highly transcribed genes in a *Δsaf5* strain.

## Discussion

Splicing, a fundamental mechanism across all eukaryotes, plays a pivotal role in the precise regulation of gene expression. Misregulation of splicing, whether stemming from mutations within pre-mRNA cis-elements or a deficiency in various splicing factors, can significantly impact cellular function. These factors may directly integrate into the spliceosome or aid in its formation and the recognition of pre-mRNA splicing sites, and their malfunction is often associated with a spectrum of human diseases [14,15].

The fission yeast *Schizosaccharomyces pombe* is an excellent model organism to study splicing regulation, where several known splicing factors have been well studied, whilst many others have a yet unknown function. With the *in vivo* screening presented here, we wanted to deepen our understanding of splicing regulation in fission yeast. Notably, in *S. pombe* the 5'SS sequences are quite degenerate, influenced by the upstream exonic sequence, potentially compromising recognition by snRNP U1. Additionally, the lack of conservation in the BP sequence renders intron splicing more sensitive to the absence of specific splicing factors, as observed with Prp2 [27]. This degeneration of consensus sequences is especially notorious when compared with budding yeast, which has only a few hundred introns, but the vast majority fit the consensus sequences. Consequently, we decided to use a reporter strain carrying mutations in the 5'SS and BP sequences, aiming to increase the sensitivity.

Among the splicing regulators identified in our screening, we focused on Cwf12 and Saf5. Cwf12 is a core component of the NTC complex whose founding member is Prp19. Although not being part of the core spliceosome, the NTC complex is associated with the splicing machinery and is required for the two catalytical splicing reactions [37]. Saf1, another candidate identified in the screen, has also been shown to interact with the NTC complex [38]. Strikingly, while the splicing of all our reporters was significantly affected in *Δcwf12* cells, yet the global impact on all fission yeast introns was comparatively modest. This observation prompted us to realize that we had engineered an artificial intron (dmut reporter) with splicing sequences divergent from the fission yeast consensus and, in the absence of Cwf12, *S. pombe* struggles to process this artificial intron efficiently.

In contrast, our findings regarding Saf5 presented opposed outcomes: while the reporter was less affected in *Δsaf5* cells when compared to *Δcwf12* cells, there was a more substantial impact shown in genome-wide scale, with numerous introns exhibiting decreased splicing efficiency. Saf5 is conserved from yeast (LOT5 in *S. cerevisiae*) to metazoans and the human ICLN gene (CLNS1A) complements the *Δsaf5* slow-growth phenotype [32]. ICLN, together with WD45 and the PRMT5 methyltransferase forms a complex termed methylosome that acts together with the SMN complex during the early steps of snRNP biogenesis [39]. The core spliceosome consists of five snRNPs and the biogenesis of these snRNPs is an ordered multistep process where ICLN acts as assembly chaperone assisting in the recruitment and the formation of the Sm core protein complex on the snRNA [40]. Although these functions have not yet

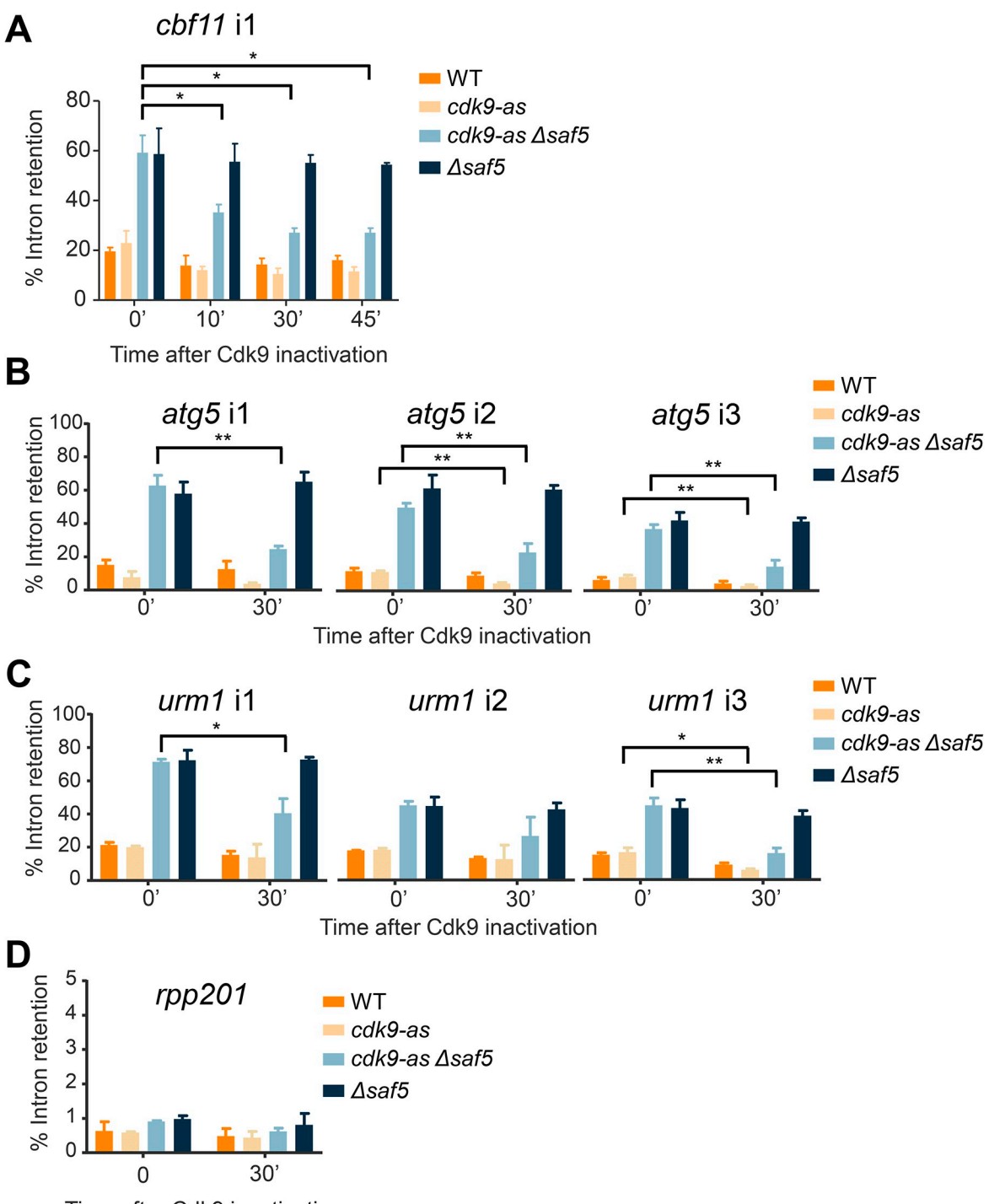

**Fig 5.** (**A**) Percentage of intron retention in *cbf11* single intron is above 50% in *Δsaf5* strain. Splicing efficiency is improved after inactivation of Cdk9, reaching similar levels of the splicing detected in WT or *cdk9-as* cells. Time in minutes in indicated at the bottom and the strains used are shown on the right. (**B-C**) Intron retention of genes with three introns, *atg5* (**B**) or *urm1* (**C**) was measured in untreated cells or in cells after 30 minutes of Cdk9 inactivation. Time in minutes in indicated at the bottom and the strains used are shown on the right. (**D**) Intron retention of the single intron from *rpp201*, a gene with full splicing efficiency in in *Δsaf5* cells, was determined after inactivation of Cdk9. Significant differences in the whole figure were calculated using a two-sided t-test (* p<0.05, ** p<0.01).

been rigorously confirmed for Saf5 in *S. pombe*, the functional complementation by the human homolog points towards a conserved role.

We have been able to show that those introns whose splicing was affected in *Δsaf5* cells were indistinguishable in terms of nucleotide sequence or composition from those unaffected. However, we successfully established a clear link between highly transcribed genes and the necessity of Saf5 for the correct splicing of their pre-mRNAs. This correlation was evident in instances where a decrease in transcriptional speed resulted in the splicing of these genes becoming largely independent of Saf5. Research spanning several decades has provided substantial evidence indicating that a significant portion of splicing occurs co-transcriptionally. Transcriptional dynamics can influence the selection of splice sites within pre-mRNA, thereby leading to alternative splicing events and the generation of diverse mRNA isoforms [41–44]. The functional interplay between transcription and splicing is proposed to be governed by spatial and kinetic mechanisms [45]. Central to both mechanisms is the carboxy-terminal domain (CTD) of RNA Polymerase II (Pol II), whose phosphorylation status regulates transcription elongation rates and facilitates physical interactions with splicing factors such as Prp40, thereby facilitating association with the U1 complex [46]. The kinetic model posits that differential Pol II elongation rates may dictate the availability of splice sites [43,45].

The data presented herein unequivocally establish a correlation between highly transcribed genes and the indispensable role of Saf5 in ensuring efficient splicing. Given its functional homology to ICLN, it is tempting to speculate that the absence of Saf5 creates a bottleneck in the biogenesis of functional snRNPs. The functional coupling of rapid transcription rates with splicing may place a heightened demand on snRNPs, with highly transcribed genes relying particularly on Saf5 for adequate snRNP supply. In scenarios where snRNP levels are insufficient, splicing and transcription may become uncoupled and fail to match the pace of transcript production. This potentially elucidates why the splicing of highly transcribed genes exhibits heightened dependence on Saf5. Future investigations will be required to validate this hypothesis and deepen our understanding of the intricate relationship between transcription, splicing, and snRNP biogenesis.

## Materials and methods

### Strains and media

All strains used are listed in S3 Table. Media were prepared as described previously [47]. Strains were transformed by the lithium acetate method, as described before [48]. Plasmids were inserted at the *leu1* locus after linearization with the restriction enzymes *NdeI* or *NruI* (New England Biolabs #R0111S and #R0192, repectively). Positive clones were selected by auxotrophy selection and nourseothricin or G418 resistance. Deletions were confirmed by PCR using the primers described in S4 Table.

### Reporter strains

The integrated fluorescence-based *in vivo* splicing reporter constructs used in this work (Fig 1A and 1B) consisted in the 170 first nucleotides of *S. pombe* intron-containing gene *rhb1*, preceded by Red Fluorescence Protein (RFP) and the HA epitope, and followed by Yellow Fluorescence Protein (YFP). The single and double mutant transcripts differ from the wild type (WT) in two mutations: one at the 5' splicing site (5'ss) and other at the branchpoint (BP) sequence. These two mutations have been previously described to affect splicing efficiency in *prp2* mutant background (*prp2-1* and *prp2-2*) at the restrictive temperature (37°C) [27]. cDNA construct was used as a read-out for full splicing.

## Generation of the reporter library

The BIONEER deletion collection library V2 of *S. pombe* contains approximately 3400 deletions of non-essential genes. A mini-collection of 41 splicing-related genes (S1 Table) was selected to check the efficiency of the reporters before crossing them with the whole BIONEER collection. The PEM2 (Pombe Epistatic Mapping) strain [49] was used to cross systematically the selected deletions arrayed in a 96-well plate, with the reporter strains JA2920 and JA2922 (JA1814 strain transformed with plasmids pAY1009 and pAY1011, respectively), as described before [50]. Spores resulting from crosses in Minimal Media minus Nitrogen (MMN) were germinated in YE5S plus Cycloheximide and G418 to eliminate diploids and haploid parental strains. Final selection was done in YE5S plus Cycloheximide, G418 and Nourseothricin.

## Flow cytometry

**Sample preparation.** Strains were cultured in 96-well plates using a 96- pin replicator (V&P Scientific) as previously described [50]. In each plate, JA2922, JA2920 and wild type (WT) strains were added with a pipette tip in empty wells (G6, H6 and F6, respectively), used as controls. When the whole collection was crossed, the procedure was performed similarly, but the control strains (JA2922, JA2920 and WT) were added in wells H2, H3 and H12, respectively.

**Acquisition.** BD LSRFortessa flow cytometer with a High Throughput Sampler (HTS) was used to measure RFP and YFP fluorescence. RFP was excited at 561 nm and detected using a 610/20 band-pass filter and 600 LP emission filter; while YFP was excited at 488 nm and detected using a 525/50 band-pass filter and 505 LP emission filter. Populations were obtained by hierarchical gating using i) forward (FSC) and side (SSC) light scattering ii) FSC-A against FSC-H iii) FITC-A and PerCP-Cy5-5-A. 80 μl of sample were mixed twice for each well at 180 μl/s. 80 μl were analyzed at 3 μl/s and washed with 800 μl of FACS Flow. 10.000 events per well were recorded. Data acquisition and processing was performed with BD FACSDiva Software 6.0.

**Data processing.** YFP/RFP (FITC/PETexas Red) ratios were calculated for each well using raw FITC and PETexas Red median values of the final population. WT strain was used as a control, by subtracting its PE-Texas Red value from each well, to eliminate autofluorescence. Minimal threshold of counted events considered for trustable results was 1000. Triplicates were performed with the mini-collection of 41 deletion strains, and the mean of each well ratio is represented. In the case of the whole collection of mutants, at least duplicates were performed. Log2 of the fold change of each well ratio regarding the WT ratio is represented.

## Splicing efficiency measurement by PCR and RT-qPCR

RNA extraction was performed as described previously [26]. For RT-PCR analysis, 25 μg of total RNA were digested with DNaseI (Roche) for 30 minutes at 30°C; DNaseI was heat inactivated at 75°C for 10 minutes. RNA (4 μg) was reverse-transcribed using oligo-dT and the Reverse Transcription System (Promega). cDNA samples were diluted 1:10 and 5 μl of each sample were used for PCR reactions (final volume of 20 μl) with primers J1907 (forward RFP sequence: CACCATCGTGGAACAGTACG) and J1752 (reverse rhb1 exon 2 sequence: ATTACCCGGGTTATGGATAATACGATTCAACG). PCR products were separated by electrophoresis on a 2% agarose gel in a 1xTBE buffer with ethidium bromide. Digital images were acquired with BioRad software. Other introns apart from the rhb1 construct were checked by PCR, and the primers used for it are listed in S4 Table. Splicing efficiency of some of the candidates was checked also by RT qPCR. Duplicates of cDNA samples and specific primers were

mixed with Lightcycler SYBR Green I Master (Roche). Reactions were carried out on a Light-cycler 480 Instrument II (Roche) for the quantitative PCR.

## Plasmid constructions

Plasmids used in this work are listed in S5 Table. The genomic region of rhb1 was amplified from WT genomic DNA. Primers used are listed in S4 Table. All PCR products were cloned into plasmid pAY856 (derived from plasmid pAY24), in which *nmt41* promoter has been changed by *sty1* promoter from plasmid pAY686. Plasmid pAY24 is derived from the plasmid pJK148 and contains the *leu1* gene for stable integration and selection into the *leu1* locus. Mutations were introduced by overlapping extension methodology previously described [51]. The sequence of all constructs was confirmed by Sanger sequencing at the Genomic Core Facility.

Plasmids containing the *mmi1* gene (with or without introns) tagged with sfGFP at the N-terminus and under the control of 1 Kb of the *mmi1* promoter were constructed by Gibson assembly of the PCR products obtained to amplify the *mmi1* promoter (primers J2974-J3044) from genomic DNA, sfGFP (primers J3045-J3046), *mmi1* ORF or *mmi1* cDNA (J3047-J3048) from genomic DNA or from cDNA, respectively, and the pAY24 plasmid digested with *PstI* and *SmaI* (New England BioLabs, #R0140 and #R0141, respectively). The sequence of all constructs was confirmed by Sanger sequencing at the Genomic Core Facility.

## Statistical analysis

Significant differences between samples were determined by the two-sided student's t-test (* $p < 0.05$, ** $p < 0.01$, *** $p < 0.001$).

## RNA sequencing and analysis

Total RNA from *S. pombe* YE5S cultures, was obtained and processed as described previously [26]. As suggested by ENCODE guidelines for RNA sequencing experiments, two biological replicates were performed and analyzed. Libraries were prepared using TrueSeq kit following manufacturer instructions. Final libraries were analyzed using Agilent DNA 1000 chip to estimate the quantity and check size distribution. Sequencing was done using the NextSeq2000, paired reads, 150 nts. Raw FASTQ files were first evaluated using quality control checks from FastQC 0.11.9. High quality reads were mapped to the *S. pombe* ASM294v2 reference using STAR. Resulting alignment files are processed to evaluate alignment quality and read distribution with Qualimap and to obtain gene read counts using HTSeq. Raw read counts were normalized with DESeq2 and employed to find differentially expressed genes between sample conditions. The correlation between biological duplicates for all RNA-seq couples according to Pearson coefficient was > 0.98. SPLICE-q [52] was used for splicing efficiency quantification.

## Growth assays

Cells were grown in YE5S at 30°C to logarithmic phase (OD600 0.5). For the spots assay, cultures were concentrated to OD600 2.5, and 1/10 serial dilutions spotted onto plates of rich media. Plates were imaged after three days of growth at 25°C, 30°C or 37°C. In the case of growth curves, logarithmic cultures were diluted to a OD600 of 0.1 and plated into 96 well plates. To monitor cell growth, optical density at 600 nm was automatically measured every hour during 48 h using a spectrophotometer.

## TCA protein extraction and Western blot

Cells were grown in YE5S at 30˚C to logarithmic phase (OD600 0.5) and total protein extracts were obtained by trichloroacetic acid TCA precipitation as described [53]). Samples were separated by SDS-PAGE and detected by immunoblotting. sfGFP-tagged proteins were visualized with monoclonal anti-GFP (Takara). Anti-Sty1 polyclonal antibody [54] was used as loading control. Both proteins were detected in the same membrane.

## Meiotic gene expression measurement by RT-qPCR

RNA extraction, DNaseI digestion and reverse transcription was performed as above. cDNA samples were diluted 1:10 and 2 μl of each sample were used for PCR reactions (final volume of 20 μl) with primers J3130 (Forward qPCR Bqt1: ACTACGACGCTTATTTCTTTTGAAC A), J3131 (Reverse qPCR Bqt1: CTCACAGTTTGAATCAGTGCATACA), J2409 (Forward qPCR Rec10: GCTTACCACATAAATTGTAACAAAG), J2410 (Reverse qPCR Bqt1: GACCA GATTAACTTCAATTTGCGCC), J3118 (Forward qPCR Meu13: GCTTTAAATAACTCACT CAGTCCAGC) and J3119 (Reverse qPCR Meu13: TCGAAGAGATTCGAGTTTTGAGCT). *tfb2* expression was used as housekeeping control with primers OLEH-842 (Forward qPCR Tfb2: CTGTTCAGGTTTTGCACTTTTTATT) and OLEH-843 (Reverse qPCR Tfb2: TTCA AGCATGATTTGTTGTGTATCT). Duplicates of cDNA samples and specific primers were mixed with Lightcycler SYBR Green I Master (Roche). Reactions were carried out on a Lightcycler 480 Instrument II (Roche).

## Supporting information

**S1 Table. List of non-essential genes included in the subcollection.**
(XLSX)

**S2 Table. List of strains isolated in the screening.**
(XLSX)

**S3 Table. List of strains used in this study.**
(XLSX)

**S4 Table. List of primers used in this study.**
(XLSX)

**S5 Table. List of plasmids used in this work.**
(XLSX)

**S1 Fig. Characterization of the reporter system.**
(TIF)

**S2 Fig. Splicing efficiency in *Δsaf5* and *Δcwf12* strains.**
(TIF)

**S3 Fig. Saf5 regulates the amount of Mmi1 in the cell.**
(TIF)

**S4 Fig. Splicing efficiency in 3-intron genes.**
(TIF)

**S5 Fig. Splicing improves in *Δsaf5* after Actinomycin D treatment.**
(TIF)

## Acknowledgments

We thank Tokio Tani, Assen Roguev, Robert Fisher and the National BioResource Project (NBRP) for different strains. We thank members of the Oxidative Stress and Cell Cycle Group for insightful comments. We are especially grateful to David Castillo for implementing a pipeline in BatchX to analyze splicing in the RNAseq experiments.

## Author Contributions

**Conceptualization:** Elena Hidalgo, Stefan Hümmer, José Ayté.

**Data curation:** Montserrat Vega.

**Formal analysis:** Sonia Borao, Montserrat Vega, Stefan Hümmer.

**Funding acquisition:** Elena Hidalgo, José Ayté.

**Investigation:** Sonia Borao, Montserrat Vega, Susanna Boronat, Stefan Hümmer.

**Resources:** José Ayté.

**Supervision:** Stefan Hümmer, José Ayté.

**Validation:** Susanna Boronat.

**Writing – original draft:** Stefan Hümmer, José Ayté.

**Writing – review & editing:** Stefan Hümmer, José Ayté.

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
