## [Decision Letter · Decision Letter 0]

19 Feb 2024

Dear Dr. Ayte:

Thank you very much for submitting your Research Article entitled 'A systematic screen identifies Saf5 as a link between splicing and transcription in fission yeast' to PLOS Genetics.

The manuscript was fully evaluated at the editorial level and by three independent peer reviewers. You will see in the enclosed and attached documents, that the reviewers appreciate your genetic and biochemical studies which characterize Saf5's role in mRNA splicing. However, the reviewers are not convinced that you have proven the link between Saf5's role in splicing and transcription. Each reviewer suggests additional types of data which would bolster or modify your model/conclusions; we believe that addition of data from any of the suggested experimental approaches would strengthen your conclusions. Further, each reviewer requests additional statistical analyses of your data. Based on the reviews, we will not be able to accept this version of the manuscript, but we would be willing to review a much-revised version. Your revisions should address the specific points made by each reviewer. We will also require a detailed list of your responses to the review comments and a description of the changes you have made in the manuscript.

If you decide to revise the manuscript for further consideration at PLOS Genetics, please aim to resubmit within the next 60 days, unless it will take extra time to address the concerns of the reviewers, in which case we would appreciate an expected resubmission date by email to plosgenetics@plos.org.

Accompanying reviewer comments and attachments are included with this email; please notify the journal office if any appear to be missing. They will also be available for download from the link below. You can use this link to log into the system when you are ready to submit a revised version, having first consulted our Submission Checklist.

To enhance the reproducibility of your results, we recommend that you deposit your laboratory protocols in protocols.io, where a protocol can be assigned its own identifier (DOI) such that it can be cited independently in the future. Read more information on sharing protocols at https://plos.org/protocols?utm_medium=editorial-email&utm_source=authorletters&utm_campaign=protocols

We are sorry that we cannot be more positive about your manuscript at this stage, but we look forward to receiving a revised version. Please do not hesitate to contact us if you have any concerns or questions.

Yours sincerely,

Anita K. Hopper

Academic Editor

PLOS Genetics

Geraldine Butler

Section Editor

PLOS Genetics

Reviewer's Responses to Questions

**Comments to the Authors:**

Reviewer #1: Borao et al.

In the submitted manuscript, the authors screen the S. pombe deletion library using fluorescence-based reporters to identify new genes with links to splicing efficiency. They identify the Cwf12 and Saf5 genes for follow up work, noting that Saf5 but not Cwf12 mutants have defects in splicing of endogenous genes transcriptome wide. Noting that genes most sensitive to Saf5 deletion correlate with high expression and impaired splicing in multiple introns of the same gene, they hypothesize that splicing defects linked to Saf5 deletion may be related to transcription elongation. This is tested using a P-TEFb/Cdk9 mutant allele that indeed suppresses splicing defects in the Saf5 deletion background. The authors conclude that Saf5 likely has functions related to splicing in a co-transcriptional context.

The manuscript is very interesting and the use of the fluorescence-based screen across an entire deletion library is impressive and appealing. The validation of the reporters in the context of the prp2-1 allele is reassuring.

Comments:

- The link between transcription elongation and suppression of Saf5 defects seems underdeveloped. The rescue in the context of a Cdk9 mutant allele is very appealing, but since Ser2 phosphorylation is linked to both transcription elongation rates and recruitment of splicing factors, it seems premature to conclude that the effects they see must be due to changes in elongation rates.

- Related to this, the authors should articulate what they mean by “transcription rate” when they refer to highly transcribed genes; they have only tested for effects on transcription elongation rates, but highly transcribed genes will also be intimately linked to transcription initiation events/rates. Ideally the authors would validate the hypothesis linking Saf5 to transcription elongation using another method that does not rely on Ser2 phosphorylation, such as a drug (Actinomycin D?) or a mutant of RNA polymerase that has slower elongation rates. Alternatively, to at least rule out an effect due to initiation rates, an intron containing reporter with an inducible promoter could be used, to show that low versus high transcription initiation rates are not linked to Saf5 associated changes in splicing efficiency.

- I would like to see the equivalent data in Figure 4F for multi-intron genes that are not highly expressed and whose expression was not affected by deletion of Saf5. I think this is an important control that would help substantiate the main hypothesis.

- The Discussion is quite brief. It would be helpful to bring up other work linking transcription elongation rates & Rbp1 CTD phosphorylation to efficiency of splicing as this literature is quite extensive. Further discussion of links to the SMN complex, its function in snRNP assembly and conservation of this process in budding yeast, fission yeast and humans would also be appropriate.

Minor issues:

There are some minor English languish issues in the manuscript which would benefit from editorial revision, including but not limited to:

- The term “life construct” is not commonplace; perhaps “integrated fluorescence-based in vivo splicing reporter construct”, or some other version of this?

- Abstract: “weakest splicing rate”; the term “rate” is typically linked to a quantitative measure of enzyme kinetics (i.e. k) which is not tested here.

-

Also:

- Some description of the Nineteen Complex would be helpful and appropriate. There are previously established links (i.e. Syf1), between the NTC complex and transcriptional elongation, and so the authors should contextualize their results in light of these. Both Cwf12 and Saf5 are indicated in the abstract as being related to the NTC (line 25) but in the text (line 171) only a link from Cwf12 to NTC is provided. Clarification needed.

- Initial work (Figure 1) is performed with integrated versions of the reporters, but were plasmid versions used for Figure 2A (line 132; “transformed/transformants”)? This should be clarified.

- The difference between how candidates were identified in 2C and how candidates were confirmed in Figure 2D is not clear. Is the assay in 2D the same as what was done in the initial screen in 2C?

Reviewer #2: Borao, Vega and colleagues report the results of a screen in S. pombe for genes important for efficient splicing. Using bichromatic reporters, they first validate their experimental set up by showing that in a Prp2 ts mutant strain, reporters harboring mutations in the 5’ splice site (or in the 5’SS and the branchpoint) show decreased splicing under conditions of partial inactivation of Prp2. A pilot screen using 41 ko strains of genes encoding factors involved in RNA metabolism identified cwf12, saf5 and saf1 as displaying a splicing defect when studied using the 5’SS or double mutant reporters. A genome-wide screen confirms the cwf12 and saf5 hits and reveals over 30 additional gene hits. The authors then focus on cwf12 and saf5. RNA-seq analyses do not detect splicing defects in the cwf12 ko strain but substantial defects in the saf5 ko strain, not associated with detectable differences in splice site sequences, A/T content or intron size. saf5 deletion had a profound effect on gene expression, particularly enhancing mRNA levels of over 200 genes, including genes normally expressed only in meiosis. Following this lead, the authors study splicing of introns in the mmi1 gene, which encodes a protein that targets mRNAs encoding meiotic genes for degradation during growth and suggest that cumulative splicing defects in each of these introns explain the effects on overall expression of the gene, with consequences on the upregulation of meiotic gene targets. They find that the genes containing the most affected introns by saf5 ko are generally more expressed. Finally, the authors tested whether decreasing RNA pol II elongation rates by Cdk9 inactivation affected the effects of the absence of saf5 and conclude that slowing down transcription relieves the inhibition of intron splicing induced by saf5 deletion.

These are interesting results that connect saf5 with splicing efficiency and transcription and describe coordinated effects on multiple introns of the same primary transcript. As the mechanisms of coupling between transcription and splicing remain an important open question, the results by Borao, Vega et al provide new information that can trigger further work on this topic.

In my opinion the following revisions could help to improve the manuscript:

1) It is not straightforward to interpret, from the scheme in Figure 1A, why the unspliced transcript should not produce YFP.

2) Figure 1C: the behavior of the cDNA construct serves as the reference for the YFP/RFP values of the other constructs in each of the two conditions, but is the expression of the cDNA identical under restrictive and under permissive conditions?

3) I would favor showing the RNA analysis of Supporting Figure 1 in Figure 2 because it convincingly confirms that the bichromatic reporter read out is linked to splicing defects.

4) Figure 2B: are the differences between wild type and mutant strains statistically significant for the WT reporter (0.7-0.9 ratios)? Deltasaf1 is not shown as statistically significant for the dmut reporter in Figure 2B but it is statistically significant in Figure 2D. And again not statistically significant in Figure 2C.

5) Lines 161-163: SPAC1705.02 is mentioned both as one of the four hits previously involved in the regulation of splicing and as one of the remaining 33 hits not related to splicing.

6) The result with mpn1 is interesting and Ref. 30 reports a function for mpn1 in U6 snRNA processing: it would be good to discuss possible mechanisms potentially explaining the hit, particularly in the light of the functions of Cwf12 in connection with U6 snRNA function.

7) Figures 3C/3D: it is not very clear what is represented in these graphs. Do Top 5% and Bottom 5% indicate introns with the highest variation compared to wild type? In that case “Splicing efficiency” in the X axis should indicate that this is a relative value. More generally, how do the 5% Top and 5% Bottom efficiently spliced introns in the WT compared with the 5% Top and Bottom efficiently spliced introns in the two mutant strains? And how do the Intron size and A/T content compare for these two categories in the three different strains?

8) I find the result of Supporting Figure 3B very relevant for the authors’ case and I would encourage them to include it in the main Figures. Why are two bands detected for sfGFP-Mmi1 in the western blot of Supporting Figure 3A?

9) The result of Figure 4E makes a point but it may be more compelling to plot splicing efficiency in deltasaf5 vs FPKMs (or use a heat map of splicing efficiencies to paint the dots of the volcano plot in Figure 4C).

10) The result of Figure 4F is clear but it would be good to see whether the converse if also true, i.e. if, in genes containing multiple introns (e.g. 3) that do not display defects in splicing of the first intron, the splicing of subsequent introns is also not affected by saf5 depletion.

11) Lines 253-254: “The addition of the bulky ATP analog barely affected intron retention (Figure 5A). However % of intron retention is decreased about 2-fold (I would argue that this is a significant change) after 10´of Cdk9 inactivation in the Cdk-as strain. (Wouldn’t a control with the wild type strain treated with 3-MB-PP1 be necessary here?) Furthermore, the effect of Cdk9 inactivation on % of intron retention in the saf5 ko strain is also about 2-fold. It could therefore be argued that Cdk9 inactivation has a similar fold effect in improving intron splicing in the presence or absence of saf5. The same argument applies to the results of Figure 5B. Could Cdk9 inactivation attenuate the splicing defects caused by the perturbation of any other splicing factor (e.g. by providing a wider time window for the spliceosome to assemble / be activated?), i.e. is the link between saf5-dependent splicing and transcription specific of saf5?

12) The Discussion is succinct and could be enriched by discussing models by which RNA polymerase elongation rates can influence the effects of saf5 depletion over other defects in RNA processing; further discussion of known functions of saf5 would be also helpful for the general reader.

Reviewer #3: Attached

**Have all data underlying the figures and results presented in the manuscript been provided?**

Reviewer #1: Yes

Reviewer #2: Yes

Reviewer #3: Yes

PLOS authors have the option to publish the peer review history of their article (what does this mean?). If published, this will include your full peer review and any attached files.

If you choose &

---

## [Decision Letter · Decision Letter 1]

11 May 2024

Dear Dr. Ayte,

Thank you very much for submitting your revised Research Article entitled 'A systematic screen identifies Saf5 as a link between splicing and transcription in fission yeast' to PLOS Genetics.The manuscript was evaluated by the same three experts who reviewed your original submission. The reviewers appreciated your revisions and agree that  the manuscript is very improved. Two of the reviewers conclude that the manuscript is now acceptable for publication in PLOS Genetics, but Reviewer 3 has remaining concerns, mainly regarding Fig. 5. These concerns seem valid and therefore we believe that they should be addressed in an additional revision.

We therefore ask you to modify the manuscript according to the review recommendations. Your revisions should address the specific points made by Reviewer 3.

1) Provide a detailed list of your responses to the reviewer's comments and a description of the changes you have made in the manuscript.

Yours sincerely,

Anita K. Hopper

Academic Editor

PLOS Genetics

Geraldine Butler

Section Editor

PLOS Genetics

Reviewer's Responses to Questions

**Comments to the Authors:**

Reviewer #1: The authors have addressed my concerns. I support its publication in the current form.

Reviewer #3: The revised article by Sonia Borao et al is greatly improved by the additional data using actinomycin to slow transcription rates as well as the expanded and thoughtful discussion section. The authors prepared a careful point-by point response to the previous reviews that is very much appreciated. Replicate experiments and statistical analysis are now clearly shown. To this reviewer though, there are still residual concerns, stemming from the primary conclusion that transcription rate is affected in Figure 5 and in supporting Figure 5. In response to the Reviewer 3’s request to quantify the change in rate of transcription (or to show that the transcription rate is changing at all) the authors merely state that they have “repeated exactly what was published by the Fisher Lab some years ago.” This seems like a historical control (and done by another lab!!) than cannot be acceptable proof that transcription rates are changing in this experiment. Likewise, there is no new data to show that actinomycin is changing transcription rates (as has been previously reported and is expected). There are other examples where the data seem to be over-stated or interpreted inconsistently:

- The authors seem to incorrectly conclude that “The constructs containing the wild type intron or a mutation in the BP showed a slightly, but consistently reduced YFP/RFP ratio” in Figure 1, when in fact there is no change seen. The only changes seen are in the 5’ss and double mutant.

- The authors state that “The synergy between the two mutations (5’ss mut and BP mut) led to an additive effect, with the primary splicing defects predominantly attributed to the 5'ss mutation.” Since synergy, by definition, is a sum greater than its parts (ie not simply additive), this statement should be corrected.

- For Figure 5A, the authors state that “Splicing also improved in the cdk9as saf5+ strain (depicted by the pale orange bars).” This reviewer is unable to discern that improvement in the figure.

- The authors added a new Figure 5D to show that splicing is not affected in an intron in which retention is less than 1%. However, is difficult to imagine that splicing could be mor efficient than 99% so this point does not seem to support the argument.

Other points.

Does the author mean “degenerate” when stating “degenerated” on Page 14?

In the following sentence “As shown, we could not observe any effect on splicing in neither case”, neither should be replaced by either.

Reviewer #4: The authors have adequately addressed the issues raised in my previous report and I recommend publication of the manuscript in PLoS Genetics.

**Have all data underlying the figures and results presented in the manuscript been provided?**

Reviewer #1: Yes

Reviewer #3: Yes

Reviewer #4: Yes

PLOS authors have the option to publish the peer review history of their article (what does this mean?). If published, this will include your full peer review and any attached files.

Reviewer #1: No

Reviewer #3: No

Reviewer #4: No

---

## [Editor Report · Decision Letter 2]

23 May 2024

Dear Dr, Ate:

We are pleased to inform you that your manuscript entitled "A systematic screen identifies Saf5 as a link between splicing and transcription in fission yeast" has been editorially accepted for publication in PLOS Genetics. Congratulations!

Yours sincerely,

Anita K. Hopper

Academic Editor

PLOS Genetics

Geraldine Butler

Section Editor

PLOS Genetics

Comments from the reviewers (if applicable):

**Data Deposition**

http://datadryad.org/submit?journalID=pgenetics&manu=PGENETICS-D-24-00049R2

**Press Queries**

---

## [Editor Report · Acceptance letter]

30 May 2024

PGENETICS-D-24-00049R2 

A systematic screen identifies Saf5 as a link between splicing and transcription in fission yeast 

Dear Dr Ayté, 

We are pleased to inform you that your manuscript entitled "A systematic screen identifies Saf5 as a link between splicing and transcription in fission yeast" has been formally accepted for publication in PLOS Genetics! Your manuscript is now with our production department and you will be notified of the publication date in due course.

With kind regards,

Olena Szabo

PLOS Genetics

On behalf of:
